

# Identification of predictive factors of the degree of adherence to the Mediterranean diet through machine-learning techniques

Alba Arceo-Vilas[1,*], Carlos Fernandez-Lozano[2,3,*], Salvador Pita[1,†], Sonia Pértega-Díaz[1] and Alejandro Pazos[2,3]

[1] Clinical Epidemiology and Biostatistics Research Group„ Instituto de Investigación Biomédica de A Coruña (INIBIC), Complexo Hospitalario Universitario de A Coruña (CHUAC), SERGAS, Universidade da Coruña, A Coruña, Spain
[2] Department of Computer Science and Information Technologies, Faculty of Computer Science, CITIC-Research Center of Information and Communication Technologies, Universidade da Coruña, A Coruña, Spain
[3] Grupo de Redes de Neuronas Artificiales y Sistemas Adaptativos. Imagen Médica y Diagnóstico Radiológico (RNASA-IMEDIR). Instituto de Investigación Biomédica de A Coruña (INIBIC). Complexo Hospitalario Universitario de A Coruña (CHUAC), SERGAS, Universidade da Coruña, A Coruña, Spain
* These authors contributed equally to this work.
† Deceased author.

## ABSTRACT

Food consumption patterns have undergone changes that in recent years have resulted in serious health problems. Studies based on the evaluation of the nutritional status have determined that the adoption of a food pattern-based primarily on a Mediterranean diet (MD) has a preventive role, as well as the ability to mitigate the negative effects of certain pathologies. A group of more than 500 adults aged over 40 years from our cohort in Northwestern Spain was surveyed. Under our experimental design, 10 experiments were run with four different machine-learning algorithms and the predictive factors most relevant to the adherence of a MD were identified. A feature selection approach was explored and under a null hypothesis test, it was concluded that only 16 measures were of relevance, suggesting the strength of this observational study. Our findings indicate that the following factors have the highest predictive value in terms of the degree of adherence to the MD: basal metabolic rate, mini nutritional assessment questionnaire total score, weight, height, bone density, waist-hip ratio, smoking habits, age, EDI-OD, circumference of the arm, activity metabolism, subscapular skinfold, subscapular circumference in cm, circumference of the waist, circumference of the calf and brachial area.

# INTRODUCTION

The economic development, urbanisation and industrialisation worldwide have changed individuals' eating habits and lifestyles, such as smoking, excessive consumption of alcohol, sedentary lifestyle and stress, leading to a nutritional transition which its principle cost in the health sector, is the appearance of non-transmissible chronic diseases.

Corresponding author
Carlos Fernandez-Lozano,
carlos.fernandez@udc.es

A consequence of the alteration of dietary patterns is what has been called 'epidemic obesity', defined by the World Health Organisation (WHO) as the first non-viral epidemic of the 21st century, with 500 million obese people worldwide (*Finucane et al., 2011*; *Krzysztoszek, Wierzejska & Zielińska, 2015*) affecting more than 50% of the adult population in Spain (*López-Sobaler et al., 2016*; *Anta et al., 2013*; *Rodriguez Rodriguez et al., 2011*).

The assessment of nutritional status of a population is one of the best indicators of the health status of the said population, being a methodology that must include three important aspects: a global assessment, a study of the dimension and a study of body composition (*Ravasco, Anderson & Mardones, 2010*). With adequate interpretation of the findings, appropriate therapeutic measures should be taken to correct deviations from normality.

In the context of nutrition and public health, the Mediterranean diet (MD) has been forged over the centuries, being characterised by cereal, olive oil, low saturated fats and meat, moderate consumption of dairy and a regular and moderate intake of wine, being a lifestyle in accordance with geographic, climatological, orographic, cultural and environmental conditions within the countries and regions that surround the Mediterranean Sea (*Pérez & Aranceta, 2011*).

There is an increasing interest in the study of the preventive role of MD and also as a treatment for various pathologies associated with chronic inflammation, such as metabolic syndrome, diabetes mellitus, cardiovascular disease (CVD), neurodegenerative diseases, breast cancer and psycho-organic deterioration, leading to greater longevity and better quality of life (*Dussaillant et al., 2016*; *Chrysohoou et al., 2004*; *Trichopoulou, 2004*; *Serra-Majem, Roman & Estruch, 2006*; *Estruch et al., 2013*; *Sofi et al., 2014*; *Della Camera et al., 2017*). Moreover, the importance of MD has also been identified as a potential element contributing to the prevention of breast cancer (*Shapira, 2017*) or in patients carrying the BRCA mutation (*Bruno et al., 2017*). In 2010; UNESCO declared this diet an Intangible Cultural Heritage of Humanity (*UNESCO, 2010*).

Numerous studies have been published over the past decades, showing the relationship between MD intake and CVD (*Martínez-González et al., 2015*; *Widmer et al., 2015*), and meta-analyses that relate it to general health status (*Sofi et al., 2014*). In the Greek cohort EPIC (European Prospective Investigation into Cancer and Nutrition Study) a 2-point increase in adherence to this diet was associated with a 33% reduction in CVD mortality (*Sofi et al., 2014*). Additionally, the analysis of a sub-cohort of 2,700 individuals over 60 years old, with a history of myocardial infarction showed that a greater adherence to MD had an 18% drop in overall mortality (*Lack et al., 2003*). Other studies have confirmed these associations, including the follow-up of a Spanish cohort of 13,600 adults with coronary heart disease. After 5 years, it was observed that two points of increase in adherence to MD were associated with a 26% decrease in coronary risk (*Trichopoulou et al., 2007*).

Eating disorders are linked to a distorted perception of one's own body image, as well as to body dissatisfaction. The importance of a study on body dissatisfaction is due to the fact that recent investigations have confirmed that alterations in body image have a causal

participation in an eating disorder, rather than being secondary to it (*Míguez Bernárdez et al., 2011*). Body image is considered a qualitative approximation to the nutritional status of the individual (*Sámano et al., 2015*) and can be determining for their nutritional management (*Martínez-González et al., 2011*).

One of the main fields of application of Machine-Learning (ML) techniques since its origins is in the field of Biomedicine, finding previously published studies in related areas such as biomedical image (*Fernandez-Lozano et al., 2016b*), characterisation of different types of carcinomas (*Kim et al., 2017*), measurement of activity in genetic networks (*Hu et al., 2016*), deformable models for image comparison (*Rodriguez et al., 2014*), gene selection, and classification of microarray data (*Díaz-Uriarte & De Andrés, 2006*), to name a few.

Moreover, due to the great versatility of ML techniques, they have been used in a wide variety of application areas, to discover hidden patterns in the datasets: identification and authentication of tequilas (*Pérez-Caballero et al., 2017*), wearable sensor data fusion (*Kanjo, Younis & Sherkat, 2018*), predicting the outcomes of organic reactions (*Skoraczyñski et al., 2017*), animal behaviour detection (*Pons, Jaen & Catala, 2017*) or to measure the visual complexity of images (*Machado et al., 2015*). In particular, ML techniques have proven to be able to uncover unimaginable relationships in very diverse fields of application, such as image or voice recognition, sentiment analysis or language translation (*Li, Li & Wu, 2015*; *De Viñaspre & Oronoz, 2015*).

The main objective of this work is the development of ML models for the prediction of the degree of adherence to the MD. To this end, information on different anthropometric and socio-demographic variables, nutritional status and self-perception of body image is used in order to identify which of the variables have a greater influence and are key in the adherence to a healthy diet such as MD, allowing our patients to improve their quality of life and to reduce the negative effects of well-known and related diseases.

Taking into account all of the above, the experimental methodology proposed in the development of this study is based on the collection and generation of data to be analysed with our cohort in Galicia (Spain), as well as on the use of ML techniques. The purpose is to extract and explain the underlying information in the data and determine which of these variables are the most important to classify people as having either a good or poor adherence to the MD. As mentioned before, there are several health benefits related to this particular food diet, especially for: chronic inflammation, metabolic syndrome, diabetes mellitus, CVD, neurodegenerative diseases, cancer and psycho-organic deterioration, moreover leading to greater longevity and better quality of life. Thus, this study is relevant for understanding how to measure the degree of adherence, in order to ensure the aforementioned benefits.

The structure of the article is as follows: in the Materials and methods section, the subjects are presented, the variables are measured for each of them. Next, the machine learning and feature selection techniques are described, along with the experimental design followed to ensure that the results are reproducible and representative of the studied

**Table 1 Population data of the municipality of Cambre (A Coruña) for the year 2012 and sample data according to age and sex.**

| Age groups | Population | | | Sample | | |
|---|---|---|---|---|---|---|
| | Total | Men | Women | Total | Men | Women |
| 40–44 | 2,465 | 1,202 (26.9%) | 1,263 (27.8%) | 33 | 19 (12.9%) | 14 (13.2%) |
| 45–49 | 2,231 | 1,110 (24.8%) | 1,121 (24.7%) | 85 | 52 (35.4%) | 33 (31.1%) |
| 50–54 | 1,763 | 857 (19.2%) | 906 (19.9%) | 54 | 32 (21.8%) | 22 (20.8%) |
| 55–59 | 1,383 | 702 (15.7%) | 681 (14.9%) | 33 | 18 (12.3%) | 15 (14.1%) |
| 60–64 | 1,170 | 598 (13.4%) | 572 (12.6%) | 48 | 26 (17.7%) | 22 (20.8%) |
| Total (40–64) | 9,012 | 4,469 | 4,543 | 253 | 147 | 106 |
| 65–69 | 1,027 | 497 (33%) | 530 (27.5%) | 94 | 57 (38%) | 37 (37%) |
| 70–74 | 688 | 337 (22.4%) | 351 (18.2%) | 77 | 46 (30.7%) | 31 (31%) |
| 75–79 | 777 | 326 (21.6%) | 451 (23.4%) | 46 | 28 (18.7%) | 18 (18%) |
| 80–84 | 511 | 198 (13.1%) | 313 (16.2%) | 24 | 12 (8%) | 12 (12%) |
| 85-more | 431 | 148 (9.8%) | 283 (14.7%) | 9 | 7 (4.7%) | 2 (2%) |
| Total (65 and more) | 3,434 | 1,506 | 1,928 | 250 | 150 | 100 |

problem. In the next section, the results are presented and discussed, and the final section of the article includes the conclusions of the work.

# MATERIALS AND METHODS

The present study was structured as follows. Initially, a population from our cohort was selected to carry out the study; the population was grouped into two categories: with high and low degree of adherence to the MD. Once the set of the population on which the study will be carried out has been identified, the information is collected from each of the users of the health system. The type of study carried out will be described below, as well as the sample size will be justified and all measurements collected will be explained in detail. Once the dataset is generated, it will be analysed with four different ML techniques and a feature selection phase will be applied for dimensionality reduction.

## Population and data description

This is an observational prevalence study, conducted in Northwestern Spain (municipality of Cambre, A Coruña, Spain), which included a randomly selected population aged 40 years and over. The sampling population consisted of individuals residing in Cambre, identified through the National Health System card census. In Spain, the National Health System has universal coverage, and almost all Spanish citizens are beneficiaries of public healthcare services.

The sample size was calculated taking into account the total population of the municipality ($n = 12,446$). After stratification by age and gender, ($n = 503$) persons were selected to participate in the study. Sample size was estimated using the single proportion formula, with 95% confidence Interval. A sample size of ($n = 503$) subjects was estimated based on an adherence to MD rate of 50%. Precision was set at 4.3% and percentage of losses at 10%. Population data is shown in Table 1.

A personal interview was arranged with each individual. After obtaining their permission and written consent, a trained nurse proceeded to the measurement of anthropometric variables and to the collection of the necessary data to cover the questionnaires. The patients who could not go to the health centre due to personal or displacement reasons and those who suffer from a cognitive impairment, making it impossible for them to perform the study, were excluded. The study received written approval from the Regional Ethics Committee for Clinical Research (code 2012/390 CEIC Galicia).

The information described below was collected from each selected subject: socio-demographic variables: age, gender, level of education, marital status and relationships of coexistence; prevalence of arterial hypertension and smoking: the systolic and diastolic blood pressure of each patient was recorded at the beginning and at the end of the visit, obtaining the prevalence of arterial hypertension; the smoking habit was recorded according to self-reported information. Anthropometric variables: the anthropometric parameters allow us to know the state of the protein and caloric reserves, besides providing guidance to the health professional about the consequences of the imbalances in these reserves.

All measurements were made during the same session, to avoid variations in the environmental or biological conditions. For the measurement of weight and size, the person was barefoot and with light clothing; an MB-201T plus Asimed scale-rod was used with an accuracy of 100 grams (weight) and 1 mm (size). BMI was obtained by means of the $\text{BMIratio} = \frac{\text{weight(kg)}}{\text{height(m}^2)}$, and grouped according to the WHO classification of $\text{BMI} < \frac{18.5 \text{ kg}}{\text{m}^2}$: low weight; 18.5–24.99 $\frac{\text{kg}}{\text{m}^2}$: normal weight; 25–29.99 $\frac{\text{kg}}{\text{m}^2}$: overweight, and $\geq 30 \frac{\text{kg}}{\text{m}^2}$: obesity.

The waist and hip circumference was measured with an inelastic tape measure with the patient standing upright, the abdomen relaxed, the upper limbs hanging at the sides, and with the feet and knees joined together. The waist circumference was measured by taking the mid-point between the lower costal margins and the iliac crests, as it is considered a risk factor for CVD when it is wider than 80 cm in women and wider than 94 cm in men, and a very high risk if it exceeds 88 cms and 102 cms, respectively (*Alberti et al., 2009*).

The hip circumference was measured as the maximum circumference around the buttocks. Based on these two values, the waist-hip ratio was calculated using the cut-off points proposed by the WHO, where normal levels of 0.8 are found in women and one in men, higher values indicating abdominal visceral obesity, which is associated with increased cardiovascular risk (*Jover, 1997*).

The calf circumference was measured in the widest section of the ankle-knee distance (cuff area) showing a good correlation with fat-free mass and muscle strength (*Rolland et al., 2003*; *Barbosa Murillo et al., 2007*; *Bonnefoy et al., 2002*). The measurement was carried out with an inextensible tape measure in cm.

Subscapular skin fold, this fold measures truncal obesity. The measurement is made one centimetre below the lower angle of the scapula, following the natural furrow of the skin.

The scapula protrudes when the arm is carefully placed behind the back and the lower angle can be located this way. The measurement of the fold will be diagonally over an angle of 45° to ensure the correct thickness measurement. The plicometer forceps should be applied 1 cm in the inferolateral position to the thumb and finger that lifts the fold.

To assess the amount of subcutaneous adipose tissue, the skin folds were measured in millimetres in the tricipital, bicipital, subscapular and suprailiac areas. A digital calliper Trim metre was used, including a double layer of skin and underlying adipose tissue, always avoiding the muscle. The tricipital skinfold was measured longitudinally, at the back of the non-dominant upper limb, at the midpoint between acromion and olecranon, with the limb relaxed, parallel to the axis of the arm; the bicipital fold was measured at the same point as the tricipital, but on the under arm.

The circumference of the arm was measured with an inextensible anthropometric tape measure in cm. The measurement was taken at the midpoint of the non-dominant arm, in the same place where the tricipital skinfold was measured and without compression with the anthropometric tape.

Once the data of the different measurements were obtained, the mid-arm muscle circumference was found, with which the skeletal muscle mass of the patients (protein compartment) was known and expressed in cm. The arm muscle area indicated that the muscle compartment was based on the brachial circumference and tricipital skinfold measurements. The fat area of the arm indicated that the patient's fatty compartment used the total brachial area and the muscular area of the arm. The Adipose Muscular Index, which evaluates the nutritional status from the adipose and muscular areas of the arm, was also calculated, being essentially applied in the assessment of obesity.

For the determination of body fat percentage by electrical bio-impedance, a Beurer and BG55 model bio-impedance metre was used, with a maximum capacity of 150 kg and a precision of 0.1% for body fat, body water and muscle percentage, and 100 g for body weight, according to the information provided by the manufacturer.

These methods are based on physical principles, such as the different ability of conduction or resistance that the tissues show to the passage of an electric current, with greater conductivity of the lean tissues than the fatty ones (*Norman et al., 2007*). Thus, by means of bio-impedance, the following values were obtained: weight, fat mass, liquid mass, muscle mass, bone density, basal metabolic rate (BMR) and activity metabolism. Data on socio-demographic variables, such as age, gender (male/female), cohabitation (with whom the live), prevalence of current smokers, ex smokers (patient stopped smoking more than 12 months before entering the study) and non-smokers were estimated. Additionally, blood pressure was recorded.

## Adherence to the MD

Consumption of a characteristic food pattern of MD is associated with numerous health benefits. These benefits are attributed to bioactive compounds that exert synergistic effects and decrease the risk for development of chronic diseases.

In order to assess the quality of dietary habits (adherence to a Mediterranean dietary pattern), the MD adherence test was used (*Estruch et al., 2018*). It is a questionnaire

consisting of fourteen quick questions that allow us to understand whether participants' usual diet can be considered as following the parameters of the MD. Each question answered affirmatively adds a point. It is considered that a person correctly follows the MD when their score is equal to or greater than nine points.

The assessment of nutritional status was determined using the Mini Nutritional Assessment (MNA) questionnaire (*Estruch et al., 2018*). It is a validated method which, through eighteen short questions, evaluates anthropometric measures, dietary habits, lifestyle, pharmacological treatments and mobility, and performs a subjective evaluation of health and nutritional status. The total value of the MNA scale is thirty points, a score <17 being considered malnutrition, there is a risk of malnutrition between 17 and 23, 5, and well nourished subjects obtain scores of 24 points and higher.

A measure of subjective weight is included by asking: 'I consider that my weight is: (A) higher than normal, (B) normal, (C) lower than normal', following the model proposed in (*Espina et al., 2001*). Based on the answer, the population is classified into three groups: 'fairly subjective weight' those who believe to be at an ideal weight, 'more subjective kilograms' for those who believe that they are overweight and 'less subjective kilograms' for those who think they weigh less than they should.

Two of the eleven sub-scales of Garner's Eating Disorder Inventory (EDI-2) (*Garner, 1998*) were used to study body image: Body Dissatisfaction (EDI-IC) and Obsession for Thinness (EDI-OD), as they evaluate aspects directly related to perceptual alterations. The body dissatisfaction sub-scale (EDI-CI) measures the dissatisfaction of the subject with the general shape of their body or with those parts of the body that most concern those with eating disorders (stomach, hips, thighs, buttocks). The thinness obsession sub-scale (EDI-O) measures concern about weight, diets and fear of getting fat.

This questionnaire was validated in Spain by (*Corral et al., 1998*). The fourteen items of these two sub-scales were mixed in the questionnaire to avoid the subjects guessing the construct being evaluated. All items were answered and corrected according to the form proposed in the questionnaire manual. The MD Adherence test was used to determine the degree of adherence to the MD, being a short specific questionnaire of fourteen items validated for the Spanish population and used by the MD Prevention Group (PREDIMED) (*Martínez-González et al., 2015*).

## Machine learning and statistical analysis

The authors tested different ML techniques for solving this problem, using cross-validation techniques to avoid over-training, while ensuring that the generalised capability of the model is the best possible, as well as different runs of the experiments to check the behaviour of the techniques. Thus, all experiments were repeated ten times to check the stability of the results and the observed deviation between the experiments was small, as shown in the results section. In particular, a tenfold cross validation was used to divide the dataset in such a way that nine random partitions were used to train and one to validate the results, each time taking a different subset for validation.

In order to compare the performance of the ML techniques, the Area Under the Receiver Operating Characteristic Curve (AUROC) was used. This is a combined

measurement which, besides being independent of the threshold used, includes both Type I and type II errors, ensuring that it is not conditioned by differences in the total number of cases of each class (*Fawcett, 2006*).

An experimental design was employed (*Fernandez-Lozano et al., 2016a*), allowing us to divide the data using a cross-validation technique which ensured that the performance results obtained, as mentioned above, were not skewed. That is they were adjusted to the data, and researchers are able to identify which of the hyper-parameters are most suitable to find the best model with each ML techniques, according to its particular hyper-parameter configuration. To this end, the programming environment R (*R Core Team, 2016*) and the package mlr (*Bischl et al., 2016*) were used, which also allowed us to perform the considered experimental design. In addition, another of the objectives pursued by this study was to find as few variables as possible that would yield a performance value as high as possible, preferably at least equal to that obtained using all available variables. This is basically a feature selection approach where the main aims are the following: avoid overfitting and improve model performance, to provide faster and more cost-effective models, and moreover to gain a deeper insight into the underlying processes that generated the data as mentioned in (*Saeys, Inza & Larrañaga, 2007*). There are three approaches in ML to perform this process and the use of a filter approximation was chosen, for its velocity and independence of the classifier (*Saeys, Inza & Larrañaga, 2007*). In general, performing this feature selection process helps to reduce inherently the present noise in such datasets.

The final step of our experimental analysis was a null hypothesis test for choosing the best model in order to ensure whether the performance of a particular ML technique is statistically better than the others or not. In our case, as there were more than two repeated measures, an ANOVA or a Friedman test should be considered. In particular, three different conditions should be checked: normality, independence and homoscedasticity. If our results fulfil the three conditions, a parametric test is applicable, and the ANOVA one should be considered, otherwise the non-parametric version, the Friedman test. Finally, a post hoc procedure had to be used in order to correct the $p$-values for multiple testing.

## Machine learning techniques for classification problems

A large number of experiments were carried out in an attempt to identify the best ML model able to solve the problem and to ensure that the results are reproducible, real and obtained under equal conditions. In addition, the search space was explored for the best possible parameters for each technique in the same way, so that all techniques could have the same possibilities of exploration across the same subsets of data and avoid the over fit that could occur. In particular, the following well-known state-of-the-art techniques were implemented: Random Forest (RF) (*Breiman, 2001*), Support Vector Machines (SVM) (*Cortes & Vapnik, 1995*; *Vapnik, 1995*), Elastic Net (ENET) (*Tibshirani, 1994*; *Zou & Hastie, 2005*) and weighted k-Nearest Neighbours (KNNs) (*Hechenbichler & Schliep, 2004*).

Random Forest (*Breiman, 2001*) is a state-of-the-art ML technique that was used in multiple domains with good results. One of its main strengths is that the results obtained are very easy to understand, it is based on very simple concepts and in general, although it is applied with little experience in the parametrisation of hyper-parameters, good results are obtained. This technique combines multiple decision trees, each of them tuned over a subset of bootstrapped data. In this way, RF combines each of the individual predictions of the decision trees into a global prediction that, in general, is more successful than any of the simple ones. Of all the possible variables in the dataset, a number were randomly selected (with replacement) and a number of trees were constructed based on the set of examples used for the training phase and obtained from the previously selected subset. When there are classification problems, it is recommended to use a square root number of the total number of variables existing in the dataset. To explore the solution space in the best way possible, in our experiments we used a parameter domain that was adjusted by a grid search and that, for a number of trees (1,000), we explored randomly selected values of variables (2–6). In addition, values that varied (1–4) according to the size of the terminal nodes of the tree were explored.

Support Vector Machines (*Cortes & Vapnik, 1995*; *Vapnik, 1995*) is also one of the ML techniques that have been commonly used in different domains in recent years and have obtained good results. In fact, along with RF, it is one of the algorithms considered state-of-the-art, easy to understand, and the results obtained are verifiable. In problems that occur during a study, the main objective of SVM is to find the hyperplane that best separates the examples between high and low degree of adherence to the MD and at the same time to maximise the distance of separation between both examples and the hyperplane. That is it attempts to find the separation hyperplane that generalises in the best possible way (*Burges, 1998*). To achieve this goal, SVM introduces a particular mathematical concept known as kernel: it is a mathematical function that allows the conversion of the input space into a higher dimension, which is used to transform a non-separable linear problem into one that is separable. There are different kernel functions, which in general could be interpreted as a measure of similarity between two objects (60), and one of the most used is Gaussian Radial Basis (RBF), because basically any surface can be obtained with this function (61). In this case, the domain of the parameters used to search for the best model consists of a grid search of two different parameters. The first one (parameter C) is directly related to the model and is used as a balance between the classification errors and the simplicity of the decision surface, while the second (gamma parameter) is the free parameter of the Gaussian function and in particular, SVM is very sensitive to changes in this parameter. For both parameters, and according to the usual practice, values were evaluated in potencies of two between −12 and 12. To better understand this technique, the following reading materials are recommended (*Burges, 1998*; *Vert, Tsuda & Schölkopf, 2004*; *Cristianini & Shawe-Taylor, 2000*).

Elastic Net (*Tibshirani, 1994*; *Zou & Hastie, 2005*) is based on lasso (penalised least squares method) and was specifically developed to solve some of the limitations encountered for this technique (56). On the one hand, a grid search was performed on two

different parameters, the alpha penalty parameter was searched (it has values in the range of 0–1) and in particular the following 0, 0.15, 0.25, 0.35, 0.5, 0.65, 0.75, 0.85 and 1. On the other hand, the best value of the lambda parameter was used, as recommended by the authors of the technique, from values less than or equal to one to negative powers of ten, in particular the following values were used: 0.0001, 0.001, 0.01, 0.1, 1.

Finally, a simple KNN (*Hechenbichler & Schliep, 2004*) assigned, through a decision rule, an unclassified example belonging to a class by frequency of occurrence to its k-most similar neighbouring examples. Then, in accordance to the distance of Minkowski for each of the examples and following the maximum accumulated kernel densities the weighted KNN are identified (*Hechenbichler & Schliep, 2004*; *Samworth, 2012*). In particular, neighbouring values of less than or equal to nine were used. Therefore, this particular and improved implementation of a KNN used kernel functions to measure the degree of similarity of the examples, as previously mentioned in the case of the SVMs.

## RESULTS

The dataset has a total of 38 variables employed to characterise the differences underlying in the data between high and low adherence to the MD. The data has been standardised using the z-score formula to have a mean equal to zero and a standard deviation equal to 1. Four different ML techniques were used to verify the results obtained, in an attempt to identify the technique that provides the best-performing results. Initially, the analysis of the complete set of study variables is carried out. It can be seen in Figs. 1A and 1B how the techniques present a fairly stable behaviour in the prediction. Even a simple a priori technique such as KNN obtains the best results of the entire experimental phase, indicating that almost all variables contain relevant information. In any case, in order to understand whether there is noise or contradictory or correlated information that may be hindering the learning process of the algorithms, a phase of dimensionality reduction will then be carried out.

Additionally, a process of feature selection was carried out to reduce the number of variables as much as possible, so that the results could remain similar without statistical differences, if not better, for those obtained using all variables. Our approach is a filter feature selection using a *T*-test to quantify the correlation between each feature and the class (high or low adherence to the MD) before the training process. Three subsets of 4, 16 and 32 features were evaluated of the original ordering according to the highest *p*-value from the *T*-test. The average AUROC results of the execution of the ten 10-fold cross-validation experiments are shown in Fig. 1.

As the number of features increases, there is a clear growing tendency in performance and obtained results in AUROC with 16 and 32 features are very close to those obtained with the full dataset. In any case, a study should be conducted on whether the differences are statistically significant between the subsets of 16, 32 variables and the full dataset to ensure that the subset with fewer features is statistically the best option. Finally, as shown in Fig. 1A, SVM is the best model in three out of the four datasets, and manages to reach values closest to 0.94 in AUROC.
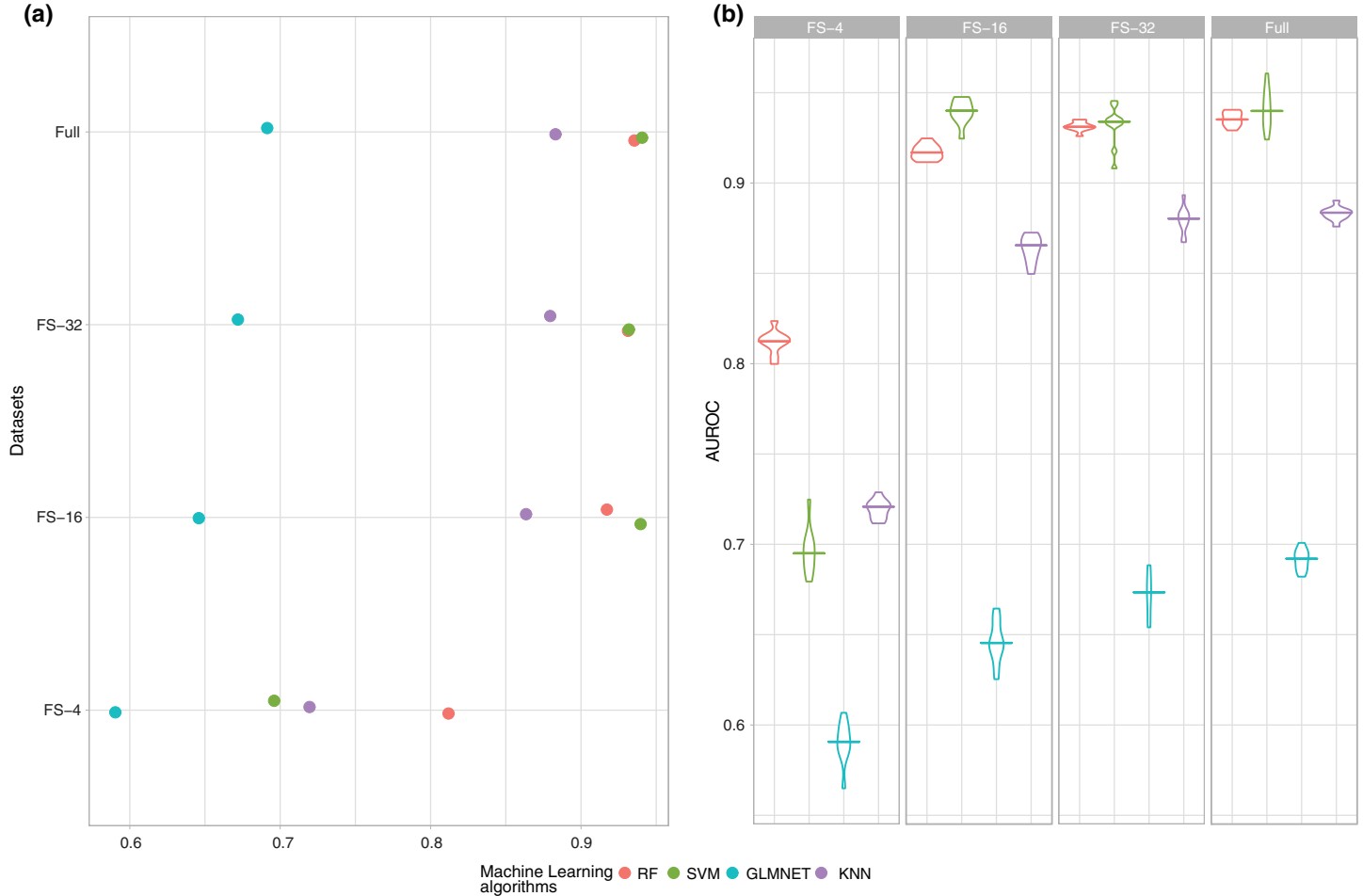

**Figure 1 Summary of the performance (AUROC) of the four ML techniques (RF, SVM, GLMNET and KNN) for each one of the subsets of features.** (A) Average of the experiments for each size analysed and (B) boxplot of the results in order to check the behaviour of the techniques through the learning process.

However, as previously mentioned, a single mean measure is not enough and it is necessary to analyse the behaviour of the models during the whole experimental phase and to verify how stable they are, as shown in Fig. 1B.

Figure 1B shows that if the number of variables is very small (4), the models are skewed and there is a higher variability in the performance because there is not enough information in the data to find a good classification model. It is also important to note that the results obtained with 16 and 32 features showed that this variability was significantly reduced until reaching average and standard deviation values very similar to those obtained using all the variables.

As observed in the two previous figures, the best results in AUROC were obtained using SVM. The same results in accuracy are shown in Fig. 2.

To check whether the difference between the three winning models (SVM with 16, 32 and all variables) is significant or not, a null hypothesis test was applied. Following the experimental methodology proposed in (*Fernandez-Lozano et al., 2016a*) for the normality

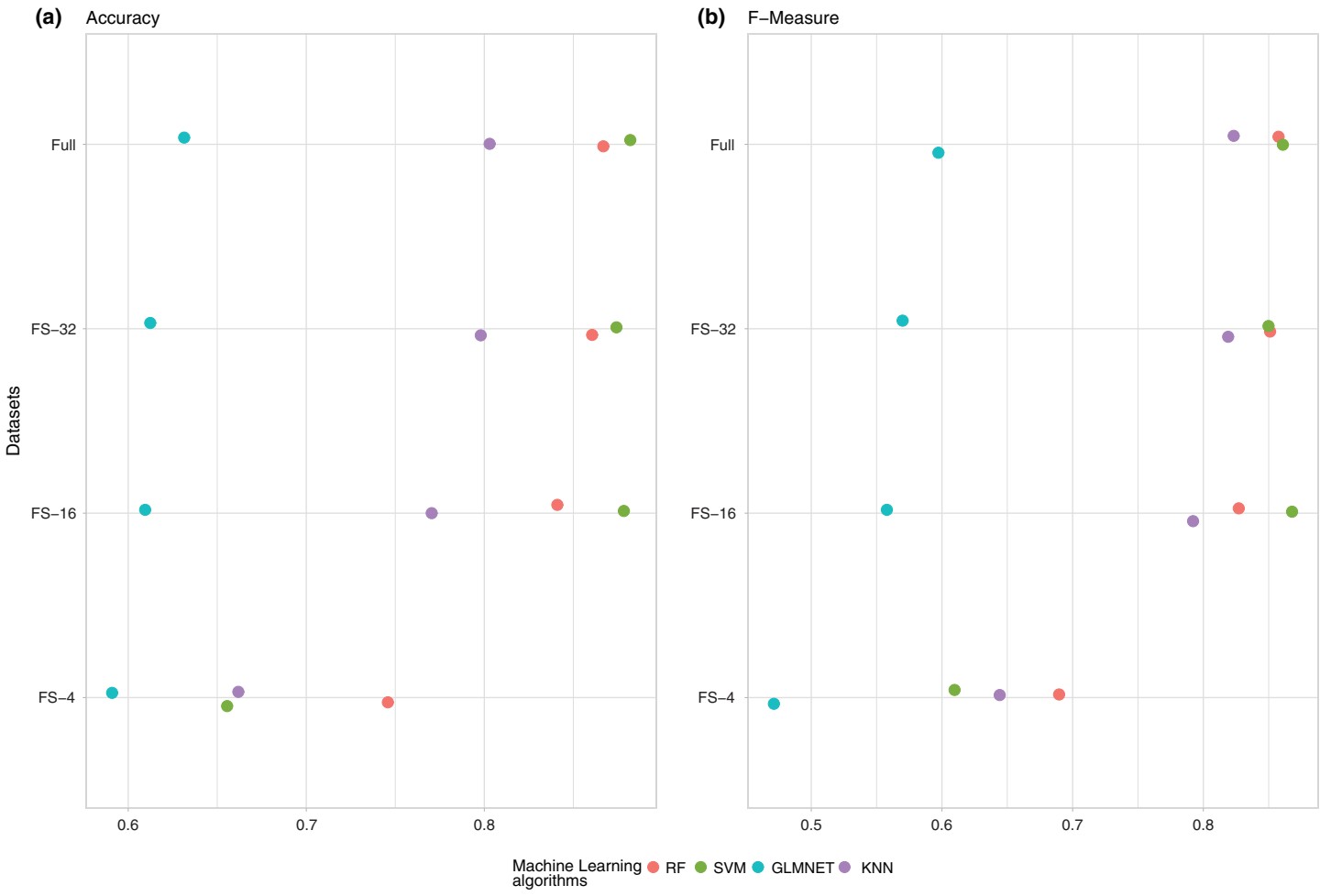

**Figure 2** Summary of the average performance of the experiments: (A) (Accuracy) and (B) (*F*-measure) of the four ML techniques (RF, SVM, GLMNET and KNN) for each one of the subsets of features.

condition we used the Shapiro–Wilk test (*Shapiro & Wilk, 1965*), with a confidence level α = 0.05. with the null hipothesis that our results follow a normal distribution. The null hypothesis was not rejected with values $W = 0.96179$ and *p*-value = 0.3438, therefore our results did follow a normal distribution.

Next, a Bartlett test (*Bartlett, 1937*) was performed, with a confidence level α = 0.05 and with the null hypothesis that our data were heterocedastic. The test result indicates that the null hypothesis should not be rejected with a value of Barlett's K-squared 2.3128 with two degrees of freedom and *p*-value = 0.3146. The result of both tests indicates that a parametric ANOVA test should be conducted, with a confidence level α = 0.05 assuming the null hypothesis that our results are statistically equal. The results of the ANOVA test indicates that we fail to reject the null hypothesis and the three ML models are statistically equal with an adjusted *p*-value = 0.1124. Consequently, a 16-feature model should be considered (BMR, MNA total score, weight, height, bone density, waist-hip ratio, smoker, age, EDI-OD, circumference of the arm, activity metabolism, subscapular skin fold,

subscapular circumference in cm, circumference of the waist, circumference of the calf, brachial area) as the best-performing one, and half of the initial features that are not relevant for the SVM were removed.

## DISCUSSION

To check whether our results are relevant and are in accordance with what has been previously published, the state-of-the-art articles published on the topic were reviewed, in an attempt to identify the degree of adherence to the variables most related to a MD. The search results led to previous studies that also found the variables identified by SVM as the most important, in particular BMR (*Cutillas et al., 2013*; *Careau, 2017*; *Srivastava et al., 2017*; *Bonfanti et al., 2014*), MNA total score (*Farias et al., 2016*; *Zaragoza Martí et al., 2015*; *Abreu-Reyes et al., 2017*), weight and height (*De la Montaña Miguélez et al., 2012*; *Buckland, Bach-faig & Majem, 2008*; *Ortega Anta & López Sobaler, 2014*; *Travé & Castroviejo, 2011*), bone density (*Romero Pérez & Rivas Velasco, 2014*; *Savanelli et al., 2017*; *Melaku et al., 2017*; *Štefan et al., 2017*) or waist-hip ratio (*Downer et al., 2016*; *Estruch et al., 2016*; *Bertoli et al., 2015*) and if the patient is a smoker (*Zaragoza Martí et al., 2015*; *Marventano et al., 2017*, *Grao-Cruces et al., 2015*). Therefore, the results were contrasted, ensuring the ability of ML techniques to identify underlying patterns in the data. According to the feature selection process, the remaining predictors are not relevant for all the ML techniques.

## CONCLUSIONS

The first model based on ML that was proposed for the prediction of the degree of adherence to the MD depended on information related to different anthropometric variables, socio-demographic variables, nutritional status and self-perception of body image.

Initially, experiments with four different ML methods were performed and feature selection techniques were applied to reduce the dimensionality of the problem. SVM is the best-performing model according to the experimental design after a null hypothesis test, and our study found that using a feature selection approach, the number of features could be drastically reduced to 16 (less than half of the initial number) achieving an equivalent performance value in AUROC. The best model obtained was an SVM with an RBF kernel as a decision function. The importance of each one of the predictors cannot be studied because a nonlinear SVM is like a black box and the internal mapping function is unknown. Furthermore, the weight vector cannot be explicitly computed.

Finally, our results are in accordance with the findings of previous publications and have primarily served to establish new factors related to the degree of adherence to the MD.

### Funding

This work is supported by the "Collaborative Project in Genomic Data Integration (CICLOGEN)" PI17/01826 funded by the Carlos III Health Institute from the Spanish National plan for Scientific and Technical Research and Innovation 2013–2016 and

the European Regional Development Funds (FEDER)—"A way to build Europe". This project was also supported by the General Directorate of Culture, Education and University Management of Xunta de Galicia (Ref. ED431G/01, ED431D 2017/16), the "Galician Network for Colorectal Cancer Research" (Ref. ED431D 2017/23), Competitive Reference Groups (Ref. ED431C 2018/49) and the European Regional Development Funds (FEDER)—"A way to build Europe". The funders had no role in study design, data collection and analysis, decision to publish, or preparation of the manuscript.

## Grant Disclosures

The following grant information was disclosed by the authors:
Collaborative Project in Genomic Data Integration (CICLOGEN): PI17/01826.
Carlos III Health Institute from the Spanish National plan for Scientific and Technical Research and Innovation 2013–2016.
European Regional Development Funds (FEDER)—"A way to build Europe".
General Directorate of Culture, Education and University Management of Xunta de Galicia: ED431G/01 and ED431D 2017/16.
Galician Network for Colorectal Cancer Research: ED431D 2017/23.
Competitive Reference Groups: ED431C 2018/49.
European Regional Development Funds (FEDER)—"A way to build Europe".

## Competing Interests

The authors declare that they have no competing interests.

## Author Contributions

- Alba Arceo-Vilas performed the experiments, analysed the data, performed the computation work, prepared figures and/or tables, authored or reviewed drafts of the paper, and approved the final draft.
- Carlos Fernandez-Lozano conceived and designed the experiments, performed the experiments, analysed the data, performed the computation work, prepared figures and/ or tables, authored or reviewed drafts of the paper, and approved the final draft.
- Salvador Pita conceived and designed the experiments, analysed the data, authored or reviewed drafts of the paper, and approved the final draft.
- Sonia Pértega-Díaz conceived and designed the experiments, analysed the data, authored or reviewed drafts of the paper, and approved the final draft.
- Alejandro Pazos conceived and designed the experiments, authored or reviewed drafts of the paper, and approved the final draft.

## Ethics

The following information was supplied relating to ethical approvals (i.e. approving body and any reference numbers):

The study received written approval from the Regional Ethics Committee for Clinical Research (code 2012/390 CEIC Galicia).

## Data Availability

Data is available at Figshare: Fernandez-Lozano, Carlos (2019): Identification of predictive factors of the degree of adherence to the Mediterranean diet through machine-learning techniques. figshare. Dataset. DOI 10.6084/m9.figshare.7628837.v2.

## Supplemental Information

Supplemental information for this article can be found online at http://dx.doi.org/10.7717/peerj-cs.287#supplemental-information.

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
