# Peer review of "Identification of predictive factors of the degree of adherence to the Mediterranean diet through machine-learning techniques"

_PeerJ Computer Science, doi:10.7717/peerj-cs.287_

## Round 0.1 · original submission · Major Revisions

Please address all the comments of the reviewers.

Please check the English grammar of your submission. It would be best to request native speakers to check through the whole paper.

Reviewer 1 ·

Basic reporting

The article uses clear and technically correct text. Sufficient background and context was provided. The structure of the article is in the standard, good form. All relevant results and explanation was included and proofs are described sufficiently. I found only a typo on page 7 in the reference to the Figure 2 (should be reference to Figure 1) and in the last line on page 7 question marks are visible instead of the reference.

Experimental design

no comment

Validity of the findings

no comment

Additional comments

The article is interesting and technically well designed. The results of this article are relevant and as authors said are in accordance with what has been previously published. Any typos was found besides two errors in references.

Reviewer 2 ·

Basic reporting

The study “Identification of predictive factors of the degree of adherence to the Mediterranean diet through machine-learning techniques” by Alba Arceo-Vilas, Carlos Fernandez-Lozano and co-authors reports the performance of 4 machine learning algorithms in classifying subject based on their adherence to the Mediterranean diet with a multi-domain feature-set comprising a total of 38 variables and 3 sub-set of 32, 16 and 4 variables.

Experimental design

1. It’s not clear which reference method was used to compare the results of the 4 algorithms. In the “Materials and Methods” section, the questionnaire used in the PREDIMED study is cited and, if this was indeed the reference, it should be specified.
2. Line 123-126. This reviewer did not understand sample size calculation.
3. One of the predictor of the 16-feature model is subscapular circumference. It is unclear what it refer to and it is not detailed in the “Materials and Methods” section.
3. The type of bio-impedance analysis (BIA) reported is not suited to research or even clinical evaluation beside the evaluation of body weight. Total-body BIA can be used used in the clinical setting for evaluation of total body water and thus estimation of fat free mass, but several assumptions are made in parameters not closely linked with impedance, limiting its applicability in research. Some degree of collinearity can be expected from measurement derived from the same measured parameters and indeed the feature selection process seems to have filtered most of BIA related parameters. Three of them remains, bone density, basal metabolic rate and activity metabolism. None of those parameter can be adequately obtained from a BIA measurement (activity metabolism is a vague definition, but to the best of this reviewer knowledge isn’t included in the parameters adequately measured or estimate by BIA). In an unsupervised fashion, it may be interesting to understand why those of all BIA measurement were related to the Mediterranean diet and not other more closely related to impedance. Nevertheless, it is the reviewer opinion that data reduction should also be based on the understanding of the methodology used and known literature data.
4. Beside BIA measurements, other parameters in the 16-parameter model seem highly related: circumference of the arm and brachial area, waist-hip ratio and circumference of the waist.
4. While performance was on par or lower than the selected model, detailing the 4, 32 and full feature model could still provide insight in the feature selection process. At minimum the full feature model should be reported.
5. Line 387. It’s not clear how the results from this study have served to “quantify the importance each factor has in the determination of the yield”. The authors report at lines 382-384 that “The best model obtained was an SVM with an RBF kernel as a decision function. The importance of each one of the predictors cannot be studied because a nonlinear SVM is like a black box and the internal mapping function is unknown. Furthermore, the weight vector cannot be explicitly computed.”
5. Some other limitations should be noted: * the PREDIMED questionnaire provide an operative definition of Mediterranean diet, food consumption recollection would have been a better reference method * no reference method were used for the assessment of body composition * no biochemical parameters were included * …
The methodological limitations in the assessment of nutritional status greatly hinders the quality of the results. If the authors are not able to address these issues, the study should only be regarded as a proof of concept on the applicability of machine learning techniques in biomedicine.

Validity of the findings

In this reviewer opinion, the application of machine learning techniques in the field of biomedicine can provide valuable insight in several disciplines, including nutrition research.

Additional comments

A value of this study is being one of the first to investigate this topic and possibly providing guidance for other researchers willing to use machine learning in future investigations.

Reviewer 3 ·

Basic reporting

This manuscript is written in a scientific and professional manner, but there are several sections where readability could be improved and some details that should be addressed.
The manuscript is self-contained and effectively addresses the initial hypothesis. Authors have provided a file with the data.
The introduction section provides enough information to contextualise the research presented in this manuscript. However, the structure of this section should be improved as there are different elements that seem to be misplaced (check comments for the authors below).
The methods, results, discussion and conclusions sections require some editing for some aspects that should be corrected.
The number of references is adequate and the references are valid.

Comments to be addressed by the authors:

• Sentence starting in line 40 reads “The assessment of nutritional status is one of the best indicators, being a methodology that must include three important aspects: a global assessment, a study of the dimension and a study of body composition Ravasco et al. (2010).” What for is this an indicator? Maybe this shouldn’t be in a new paragraph but just continuing the previous one.
• Paragraphs between lines 69 and 76 seem are difficult to contextualise in their current location. These two paragraphs seem to be disconnected from the sections above and below (the construction of the sentence implies that there is some relationship with something that has been previously stated but it is not the case here.
• Section between lines 77 and 104 seems to be strangely arranged as authors introduce the work conducted in this manuscript followed by a section describing machine learning techniques (ML) (lines 86-98) and then return to describe the objectives of this work. The logical structure would probably be the introduction of the ML techniques followed by the aims and description of the methodology applied in this manuscript
• Sentence starting in line 114 is confusing, it should be revisited.
• Paragraph starting in line 221-227 is difficult to read and confusing, what do they mean in this context with reducing the bias in the dataset? Are they referring to the use of different feature selection techniques? This should be clarified. Equally there is some redundancy in the explanation of the cross-validation approach used.

• There are problems with figure numbering. There is no reference in the text to figure 1. Probably line 325 refers to figure 1.
• Figure captions should be better explained and should contain a better description of the contents of the figure and should provide explanations about what are the different axis in the figures.
• Lines 334 and 335 refer to figure 2b but they probably refer to figure 1. This should be corrected.
• In line 341 there is a call to figure ??
• Line 345 needs to be corrected as it does not make sense.
• References are not properly formatted in the text as they are not fully enclosed between parenthesis, in the reviewed manuscript only the year was between parenthesis.
• There are a large number of references that are in Spanish. It would benefit the manuscript if the authors could provide supporting evidence in the literature in English.

Experimental design

The paper is well designed and effectively develops and applies different machine learning techniques to assess the initial hypothesis.
The methodology is well described and should suffice to replicate the results.
The methods described in the manuscript are rigorously applied.
There are however some aspects that should be addressed:
• In line 187 what was the criteria used to define the ex-smokers?
• In line 253 authors mention that there is a multiple testing correction. They should state the methodology used for the correction and should refer later one to adj. p-values in the manuscript if they are presenting adjusted/corrected p-values.
• In line 315 authors refer that variables were “scaled to the example” What did they mean? Were they trying to explain that they standardised the data using a z-score? - This should be clarified.
• Results of the Shapiro-Wilk test should be included. (line 345)
• What was the confidence level used for the Bartlett test? (line 348)

Validity of the findings

The conclusions are well supported with the data and methodology described in the manuscript.
There are some comments that should be addressed:

• In line 353, there is a conceptual error as Null hypothesis are not accepted they cannot be rejected which is different.

Reviewer 4 ·

Basic reporting

This paper has a potential to be accepted, but some important points have to be clarified or fixed before we can proceed and a positive action can be taken.We here summarize this points:1. It is really unclear to me the why authors take into consideration only few factor in study of mediterranean diet through machine-learning . This is an important point before we can start to reason about the proposed method seriously. We need to understand precisely if the proposed mechanism is able to face with the following:-different origin may have different food intake ranges and different medication depending on the other factor of body type

Experimental design

Give details regarding follow-up examinations. The authors used only 14 question for survey. Was any other past medical history obtained from two population group?
why use small data size ? did you try any other kernel beside RBF?

Validity of the findings

The authors stated there was no higher statistically significant with comparison previous such study . The authors should state the exact number of factors which significant for mediterranean diet. The author should give clear idea for factor which consider as positive aspect and which consider as negative aspects

Additional comments

The manuscript was poorly written and many grammatical error.The study must use large data size when you apply RBF kernel. The study must provide previous study examples of diet factor and compare the data.

---

## Round 0.2 · Minor Revisions

Please address the remaining comments from the reviewers.

Reviewer 3 ·

Basic reporting

This new version of the article has successfully addressed the issues previously identified by this reviewer.
The new text included in the manuscript has improved it however there are aspects that need to be addressed as there are sections that remain difficult to read or confusing.
There are changes that did not appear highlighted in the text

Experimental design

The experimental design has been improved providing some relevant additional data such as the power of the analyses fo the given sample that was analysed and extended explanations about the results of some of the statistical analyses used throughout the manuscript.

As it was stated in the previous review the paper was well designed and contains enough information to reproduce it.

Validity of the findings

No comment

Additional comments

The explanation for the calculations of the sample size needs to be clarified, not as much in the methodology, although including the formula used would be a great addition, as in its explanation as it is confusing for the reader.

The explanation about the measurement of the subscapular skin fold is repeated (lines 163-168) and then the old explanation remains at 174-176.

Placement of the new text for the adherence to the Mediterranean diet could be improved. It would make more sense if it follows lone 206, as it provides additional information about how the test is built. In this regard there is some confusing information in the new text as line 227 mentions 16 items but in the rest of the text lines 205 and 230 refers to 14 items. Which is correct?

There should be coherence in the use of digits or words for numbers (i.e. in line 205 the use of “14” and in line 220 “fourteen”)

English grammar should be thoroughly revised as there are some errors throughout the text.

Reviewer 4 ·

Basic reporting

The paper is well written. The Introduction and Background sections provide useful information for the readers.

Experimental design

Such novel kind of experiments are practically possible for predictive factors of the degree of adherence to the Mediterranean diet. Even the multi city datasets, such as the ongoing patients based nutrients intake destining project, are far from meeting the number of nutrients requirement patients needed for a congestion control study.

Validity of the findings

some information presented is need to be rectify. For example, in table.1, the authors made two major group one is below 65 and above 65, in two groups data must shows with percentage.

Additional comments

The paper entitled "Identification of predictive factors of the degree of adherence to the Mediterranean diet through
machine-learning techniques" studies degree of adherence to the Mediterranean diet with help of SVM model to check the performance, Also identify role for RBF kernel as a precise decision function in predication.It also proposes a new algorithm view to asses Mediterranean diet with machine learning methods.

---

## Round 0.3 · accepted · Accept

Thanks for your manuscript.